# HER2-Positive Gastric Cancer: The Role of Immunotherapy and Novel Therapeutic Strategies

**DOI:** 10.3390/ijms241411403

**Published:** 2023-07-13

**Authors:** Anna Pous, Lucía Notario, Cinta Hierro, Laura Layos, Cristina Bugés

**Affiliations:** 1Department of Medical Oncology, Institut Català d’Oncologia (ICO) Badalona, 08916 Badalona, Spain; apousb@iconcologia.net (A.P.); lnotario@iconcologia.net (L.N.); chierro@iconcologia.net (C.H.); llayos@iconcologia.net (L.L.); 2Badalona Applied Research Group in Oncology (B-ARGO), 08916 Badalona, Spain

**Keywords:** gastric cancer, HER2, immunotherapy, PD-1/PD-L1, antibody-drug conjugate, chemotherapy, trastuzumab, novel therapies, clinical trials

## Abstract

Gastric cancer is an aggressive disease with increasing global incidence in recent years. Human epidermal growth receptor 2 (HER2) is overexpressed in approximately 10–20% of gastric cancers. The implementation of targeted therapy against HER2 as part of the standard of care treatment in metastatic disease has improved the prognosis of this subset of patients. However, gastric cancer still has high mortality rates and urgently requires new treatment strategies. The combination of immunotherapy with HER2-targeted therapies has shown synergistic effects in preclinical models, this being the rationale behind exploring this combination in clinical trials in locally advanced and metastatic settings. Additionally, the irruption of antibody–drug conjugates and other novel HER2-targeted agents has led to the development of numerous clinical trials showing promising results. This review presents the molecular mechanisms supporting the use of HER2-targeted drugs in combination with immunotherapy and provides an overview of the therapeutic scenario of HER2-positive disease. We focus on the role of immunotherapy but also summarize emerging therapies and combinations under clinical research that may change the standard treatment in HER-2 positive disease in the future.

## 1. Introduction

Gastric cancer is the fourth leading cause of cancer death worldwide and the fifth most common malignant tumor [1,2].

Its occurrence varies markedly across different geographic regions [3], with Eastern Asian countries having the highest incidence rates, particularly China (accounting for half of all cases globally), Japan, and Korea [1].

Several environmental and genetic factors are related to the development of gastric cancer [4,5]. Most cases are sporadic, but 5% to 10% have a family history of gastric cancer [6]. Some of the most relevant risk factors include diet (especially high-salt foods, low vitamin A and C diets, and smoked or cured foods), alcohol consumption, smoking, high body mass index, gastroesophageal reflux, and Helicobacter pylori and Epstein–Barr virus infections [5].

At early stages, gastric cancer is often asymptomatic, with symptoms usually appearing when the disease is in an advanced stage [7]. Common onset symptoms include non-specific weight loss, persistent abdominal pain, dysphagia, hematemesis, fatigue, anorexia, nausea, early satiety, iron deficiency anemia, and dyspepsia [6,7].

The diagnosis requires a multimodal staging approach. Upper gastrointestinal endoscopy has been established as the gold standard for the diagnosis [5,8]. Multiple biopsies should be carried out for histological and molecular interpretation [7,9,10]. Computed tomography (CT) of the thorax, abdomen, and pelvis is the first staging modality, mostly because of its broad availability and proper accuracy [5,11]. Furthermore, [18F]2-Fluoro-2-deoxy-d-glucose (FDG) positron emission tomography (PET)–CT imaging may improve staging by detecting involved lymph nodes or metastatic disease; however, this is not routinely recommended [7]. Initial staging should also include physical examination, full and differential blood count, and liver and renal function tests [7]. Endoscopic ultrasonography is also a helpful tool for an optimized local staging [7,12]. Diagnostic/staging laparoscopy and peritoneal washings for cytology are recommended for patients with resectable gastric cancer who are also candidates for perioperative chemotherapy [7].

The most commonly used classification of gastric cancer is the Lauren classification from 1965. According to this division, three subtypes are displayed: intestinal, diffuse, and indeterminate [5,13,14].

The most frequent is the intestinal type, characterized by visible glands and cohesion between tumor cells [5,6]. This subtype is often related to environmental factors such as Helicobacter pylori infection, smoking, and dietary factors [4,8,15,16,17,18]. A mechanism of carcinogenesis of the intestinal subtype describes a progression from chronic gastritis leading to a loss of parietal cells, causing a reduction in acid secretion which induces chronic atrophic gastritis. This condition causes compensatory hypergastrinemia, which induces chronic inflammation resulting in intestinal metaplasia, dysplasia, and, eventually, adenocarcinoma [6].

The diffuse subtype is characterized by poorly differentiated and discohesive tumor cells diffusely infiltrating the gastric wall, showing a signet-ring or non-signet-ring morphology. This subtype presents a higher incidence among females and younger patients, is more prevalent in low-risk areas, and is mostly associated with heritable genetic abnormalities [5,18]. This subtype of gastric cancer originates from the normal gastric mucosa, with no clear precancerous lesions [5,15].

Gastric cancer can also be classified according to the World Health Organization (WHO) guidelines (2010), which divide the disease in several subtypes: adenocarcinoma (tubular, mucinous, and papillary), signet-ring-cell carcinoma and other poorly differentiated carcinomas, mixed carcinoma, and other less frequent gastric cancer subtypes [13].

Radical surgery remains the mainstay of curative treatment for patients with resectable gastric cancer. R0 resection with gastrectomy plus D2 lymphadenectomy is the standard of care (SoC) [19]. Unfortunately, less than 25% of patients diagnosed can be considered for resection since gastric cancer is most frequently discovered in advanced stages [2].

Despite optimal surgical treatment, there is a high rate of tumor recurrence with approximately 60% of patients presenting tumor relapse [2].

In Western countries, the reported five-year survival rate of patients treated with perioperative strategies and surgery is approximately 35 to 45% and between 0 to 10% in patients with metastatic disease, with a median overall survival (OS) of 12 months at most [2,19,20].

Considering this discouraging panorama, new treatment strategies are certainly warranted to further improve not only survival outcomes but also cancer-specific symptoms, functional status, and emotional wellness of gastric-cancer patients, as these represent key factors in their quality of life [21].

Over the last decade, significant progress has been made in better understanding the biology of gastric cancer via the investigation of its molecular characteristics. Molecular characterization is of vital relevance, given that advances in the knowledge of the molecular pathways involved in pathogenesis may allow the detection of new biomarkers capable of selecting patients eligible for targeted therapies.

## 2. Molecular Subtypes in Gastric Cancer

Gastric cancer is a heterogenous disease [22]. Traditional morphological classifications by Lauren and the World Health Organization (WHO) have limitations that do not reflect all the molecular complexity.

Several methods have been used to classify gastric cancer into molecular subtypes: next-generation sequencing (NGS) including deoxyribonucleic acid (DNA) sequencing, ribonucleic acid (RNA) sequencing, whole-exome sequencing, copy number variation analysis, and DNA methylation arrays. All these methods afford more detailed information than classic histopathological characteristics.

Molecular classification might increase the complexity and costs of diagnosis; however, it provides valuable information necessary to select targeted treatment that may increase the survival of gastric cancer patients [23,24].

### 2.1. The Cancer Genome Atlas (TCGA) Subtypes

The most comprehensive molecular characterization of gastric adenocarcinoma was reported by the TCGA (The Cancer Genome Atlas) Research Network in 2014 [22].

The study proposed a molecular classification into four molecular subtypes after analyzing 295 resected gastric tumor samples: Epstein–Barr virus (EBV)-positive (representing 9%), microsatellite unstable tumors (MSI) (22%), genomically stable (GS) tumors (20%), and tumors with chromosomal instability (CIN) (50%) [22,25,26].

EBV-positive gastric cancer is due to infection by the Epstein–Barr virus. These tumors are more frequently found in male patients and are mainly located in the gastric fundus or body. This subtype presents a high EBV burden, recurrent mutations in AT-rich interactive domain-containing protein 1A (*ARID1A*), phosphatidylinositol 3-kinase (*PIK3CA*), extreme DNA hypermethylation, B-cell lymphoma 6 corepressor (*BCOR*) mutations, amplification of Janus-associated kinase 2 (*JAK2*) and Erb-B2 receptor tyrosine kinase 2 (*ERBB2*), and programmed death ligand-1/2 (PD-L1/2) overexpression. These findings suggest a potential therapeutic role for PIK3CA and JAK2 inhibitors and immune checkpoint antagonists [22].

MSI gastric cancers are mostly due to promoter methylation, which can lead to transcriptional silencing of the DNA mismatch repair gene *MLH1*, resulting in a form of genomic instability. These tumors have a slightly higher prevalence in female and older patients (median age 72 years). The most common location is the gastric antrum. This subtype is characterized by elevated mutation rates, including mutations in *PIK3CA*, *ARID1A*, epithelial growth factor receptor (*EGFR*), *ERBB3*, and *TP53*. It also presents high levels of PD-L1 expression. Frequent frameshift mutations in repeat DNA tracts cause inactivating mutations of key tumor suppressor genes (TSGs), or frequent missense-activating mutations in oncogenes [22]. In recent years, immune checkpoint inhibitors have demonstrated relevant antitumor efficacy in this molecular subtype and currently constitute a mainstay of treatment in this population.

GS gastric cancers are more frequently diagnosed at an earlier age (median 59 years). Recurrent mutations in E-cadherin (*CDH1*) and Ras homolog family member A (*RHOA*) and CLDN18-ARHGAP rearrangements were observed. These genetic alterations may enhance invasiveness and disrupt intercellular cohesion, leading to more diffuse histologies, which confers more aggressiveness to this molecular subtype [22].

CIN gastric cancers are more frequently located in the gastroesophageal junction/cardia and exhibit intestinal histology. These tumors show marked aneuploidy, focal activation of the receptor tyrosine kinases-Ras (RTK/RAS) pathway, high frequency of *TP53* mutations, amplification of cyclins E1, D1 (*CCNE1*, *CCND1*), and cell division protein kinase 6 (*CDK6*). Amplification of *ERBB2*, *KRAS/NRAS*, *EGFR*, *ERBB3*, *FGFR2*, and *MET* are also observed [22].

There are no differences in survival outcomes between the four molecular subtypes [22]. Retrospective studies show that patients with the GS subtype had the least survival benefit with adjuvant chemotherapy and the CIN subtype had the greatest survival benefit [27].

It is of note that samples from the TCGA study were collected from resected gastric tumors. Data regarding the molecular characterization of metastatic disease are scarce. In 2021, an exploratory analysis was performed using samples from three different randomized clinical trials (KEYNOTE-059, KEYNOTE-061, and KEYNOTE-062). The analysis concluded that, in patients with advanced gastric cancer, the EBV and MSI subtypes exhibited a lower prevalence compared with the TCGA dataset [28].

### 2.2. Asian Cancer Research Group (ACRG) Subtypes

In 2015, the ACRG study proposed a new molecular classification of gastric cancer after analyzing the mRNA expression level of 300 resected tumors [29].

The study suggested dividing gastric cancer into four subtypes: MSI-high (23%), microsatellite stable/epithelial-mesenchymal transition (MSS/EMT) (15%), microsatellite stable/epithelial/TP53 intact (MSS/TP53+, p53 active) (26%), and microsatellite stable/epithelial/TP53 loss (MSS/TP53-, p53 inactive) (36%). Each of these molecular subtypes is associated with a different prognosis [29].

MSI-high cancer is located more often in the antrum and mainly exhibits intestinal histology. It is associated with hypermutation in the *ARID1A* gene, the PI3K-PTEN-mTOR pathway, *KRAS*, and *ALK*. It had the best prognosis and lowest recurrence frequency [29].

The MSS/EMT subtype is mostly diagnosed in younger patients at advanced stages. These tumors exhibit mainly diffuse histology and include a large set of signet-ring-cell carcinomas, loss of CDH1 expression, and a number of mutations. It has the worst prognosis and the highest recurrence frequency [29].

MSS/TP53+ has a higher number of mutations in *KRAS*, *SMAD4*, *ARID1A*, *PIK3CA*, and *APC* compared with the MSS/TP53- subtype; in addition, EBV infection is more frequently observed. After the MSI subtype, it has the second-best prognosis [29].

The MSS/TP53-subtype presents recurrent amplifications in *EGFR*, *CCNE1*, *ROBO2*, *MDM2*, *CCND1*, *GATA6*, *MYC*, and *ERBB2*. It has the highest prevalence of *TP53* mutations [29].

### 2.3. Comparison between Classifications

When comparing TCGA and ACRG classifications, the TCGA subtypes EBV, MSI, GS, and CIN mainly correspond to the ACRG subtypes MSS/TP53+, MSI, MSS/EMT, and MSS/TP53-, respectively. However, there are some differences between the two classifications, with a partial overlap of certain subtype characteristics, probably explained by the different patient populations, tumor sampling, and technological platforms [26].

Some of the most remarkable differences between classifications include *CDH1* mutations (more commonly detected in the GS subtype [37%] compared with the MSS/EMT subtype [2.8%]) and *RHOA* mutations (characteristic of the GS subtype and observed in the MSS/TP53+ and MSS/TP53- subtypes, yet rarely seen in MSS/EMT). Finally, the CIN and GS subtypes are distributed across all ACRG subtypes.

*HER2* gene amplifications are observed in several subtypes including CIN, GS, and EBV; however, they are more commonly associated with the CIN molecular subtype [22,25,29,30].

In the ACRG classification, recurrent focal amplifications in *HER2* were more commonly detected in the MSS/TP53- subgroup [25,29].

Based on these molecular classifications, HER2-positive gastric cancer has been more frequently associated with the CIN and MSS/TP53- molecular subtypes.

## 3. Human Epidermal Growth Factor Receptor 2 (HER2)

Human epidermal growth factor receptor 2 (HER2) is a transmembrane tyrosine kinase receptor involved in the pathogenesis and outcomes of several types of cancer, including advanced gastroesophageal adenocarcinomas [31]. The HER2 receptor belongs to the epidermal growth factor receptor (EGFR) family which consists of four members: EGFR, HER2, HER3, and HER4 [32,33].

HER2 expression is most commonly determined by immunohistochemistry (IHC) and/or fluorescence in situ hybridization (ISH), although other methodologies are available. Overexpression of the HER2 protein in gastric cancer was first described in 1986 via IHC [34].

HER2 protein expression may be classified as 0, 1+, 2+, or 3+ by IHC depending upon the extent and pattern of staining. HER2 overexpression is currently defined as IHC 3+, or IHC 2+ along with *HER2* gene amplification by ISH (chromosome enumeration probe (CEP) 17 ratio ≥ 2 (ISH positive)) [35]. However, IHC classification presents limitations, mainly due to remarkable intratumoral heterogeneity, the possibility of incomplete staining of gastric cancer cells, and inter-pathologist variability associated with the subjective interpretation of the results [36].

HER2 expression has been associated with gastroesophageal adenocarcinomas, with several studies reporting *HER2* amplification rates varying from 12% to 27% and HER2 overexpression from 9% to 23% [19,25,37,38,39,40]. Its expression is more frequent in the proximal stomach, including the esophageal–gastric junction, than in the distal stomach [40].

Some studies could not demonstrate the prognostic properties of HER2; however, a larger number of studies indicated that HER2 expression confers a more aggressive biological behavior and higher recurrence frequencies in HER2-positive tumors [40,41,42,43,44,45,46,47,48].

HER2 status is the most studied target in gastroesophageal cancer and has key clinical implications in the management of the disease [49].

Trastuzumab, the first monoclonal antibody against the HER2 receptor, was approved for clinical use, in combination with chemotherapy, as first-line treatment in patients with HER2-positive unresectable or metastatic gastric cancer in 2010 [50], representing a paradigm shift in the management of HER2-positive gastric cancer disease.

Therefore, HER2 testing is strongly recommended for all patients at the time of diagnosis, especially in the metastatic setting, due to its clinical implications [19].

Dimerization of the HER2 receptor activates two major intracellular signaling pathways: the mitogen-activated protein kinase (MAPK) (Ras/Raf/MEK/ERK) and phosphatidylinositol 3’-kinase (PI3K)/Akt, inducing cell-cycle progression, proliferation, and survival [51,52]. HER2-directed antibodies present different binding extracellular domains of the HER2 receptor. Trastuzumab, margetuxumab, trastuzumab-emtansine (T-DM1) and trastuzumab-deruxtecan (T-DXd) bind to the extracellular domain IV of HER2. Pertuzumab binds to the extracellular domain II and Zanidatamab binds to both extracellular domains II and IV of HER2. A representation of molecular activation pathways and binding union sites of HER2-directed antibodies is displayed in Figure 1.

Advances in the knowledge of gastric cancer, including the role of the HER2 pathway, the characterization of its molecular development, and a deeper understanding of the tumor microenvironment, have raised new hypotheses to improve new therapeutic strategies, including immunotherapy.

## 4. PD-L1 Expression in HER2-Positive Gastric Cancer

Several studies have shown that the immune system plays a key role in the growth of malignant tumors [53,54].

PD-L1 is an inhibitory molecule expressed in a broad range of cancers; it is a ligand of the PD-1 receptor expressed on the T cell surface [54].

The function of PD-1 is to downmodulate undesirable immune responses, and it has been shown to negatively regulate antigen receptor signaling by interacting with its ligand [55,56].

When PD-L1 is expressed on a tumor cell membrane, it interacts with the PD-1 receptor on the T cell. This interaction blocks T-cell proliferation and activity against the tumor, which allows cancer to escape from the host’s antitumor immunity [57,58,59]. Additionally, the expression of PD-L1 on tumor cells leads to the apoptosis of specific CD8+ cytotoxic lymphocytes, which further decreases the antitumor immune response [60]. These characteristics make PD-L1/PD-1 a potential treatment target in a wide range of malignant tumors, including gastroesophageal cancer.

The PD-L1 combined positive score (CPS) has been increasingly developed as a possible predictive biomarker of response to immunotherapy, and is being regularly used as a stratification marker in clinical trials [61]. This predictive score has been defined as the number of PD-L1-positive cells (tumor cells, macrophages, and lymphocytes), divided by the total number of tumor cells and multiplied by 100. The score varies between CPS ≥ 1, CPS ≥ 5, and CPS ≥ 10. In gastric cancer, PD-L1 CPS ≥ 1 generally defines PD-L1-positive tumors [62].

Some studies have reported significantly higher PD-L1 expression rates in HER2-negative tumors [24,63], while other studies reported opposite findings [64,65] or found no differences in PD-L1 expression between HER2-positive and negative tumors [7,66]. These discrepancies may be due to the heterogeneity of the populations studied, differences in the scoring methods, or in the monoclonal antibodies used for detection.

There may also be limitations in detecting PD-L1 expression, as PD-L1 can be present not only in the cell membrane but also in and outside cells. The known PD-L1 formats include membrane PD-L1 (mPD-L1) [67,68,69], cytoplasm PD-L1 (cPDL1) [69,70], nuclear PD-L1 (nPD-L1) [71,72], and serum PD-L1 (sPD-L1).

Many studies suggest that the broad distribution of PD-L1 can lead to cytotoxic T lymphocyte deactivation [73], therefore affecting cancer immunity.

Due to its wide distribution and the functions of these PD-L1 formats, the efficacy of immunotherapy may be influenced, as the antibody can present limitations in recognizing other intracellular PD-L1 formats, such as cPD-L1 and nPD-L1.

On the other hand, PD-L1 is a dynamic marker that can be expressed constitutively (driven by endogenous oncogenic pathways) or in an inducible fashion (motivated by exogenous signals) [74].

Constitutive PD-L1 is mainly regulated via MAPK (Ras/Raf/MEK/ERK) and PI3K/Akt pathways [75,76]. As previously mentioned, both molecular routes are also involved in the HER2 intracellular signaling pathway. Inhibiting these routes can regulate PD-L1 expression, which can modify the efficacy of cancer treatments.

Inducible PD-L1 is regulated by extracellular signals such as cytokines, epidermal growth factors, and hypoxic conditions. Interferon-γ (IFN-γ) is a cytokine which is known to regulate PD-L1 expression [77]. Moreover, the activation of nuclear factor-κβ (NF-κβ) has also demonstrated to have a key role in the PD-L1 expression induced by IFN-γ and tumor necrosis factor-α (TNF-α) [78].

The relationship between molecular subtypes of gastric cancer and PD-L1, HER2, and combined HER2 and PD-L1 expression, requires further investigation.

### 4.1. Trastuzumab as an Inducer of Immunity

There is strong evidence based on preclinical and clinical studies that the immune system contributes significantly to the therapeutic effects of trastuzumab in solid tumors [79,80].

The precise mechanism by which trastuzumab acts in cancer cells is not completely understood. Trastuzumab seems not only to prevent the dimerization of HER2 with other HER family members and stimulate endocytosis (HER2 internalization), it also appears to play an important role in the tumor microenvironment, implying the feasibility of immunotherapy in HER2-positive malignancies.

HER2-positive cancers have high levels of T-cell infiltration [81]. Evidence indicates an association between the immune signature/extent of tumor-infiltrating lymphocytes and the response to trastuzumab [82].

Several preclinical studies have shown that trastuzumab increases T-cell activation (antibody-dependent cellular cytotoxicity [ADCC]), recruitment of natural killer (NK) cells (degranulation and cytotoxicity), and cross-presentation by dendritic cells, inhibits angiogenesis, induces the expression of tumor-infiltrating lymphocytes, and modulates the expression of the major histocompatibility complex class II, resulting in the enhancement of cell-mediated antitumor immunity [40,62,83,84,85,86,87]. Trastuzumab also induces the secretion of inflammatory cytokines, including IFN-γ [88,89], which, as previously mentioned, upregulates PD-L1 expression [77]. This upregulation of PD-L1 has been described as a mechanism of resistance to trastuzumab [81,85,88,90]. These effects are represented in Figure 2.

Specifically in breast cancer, Chaganty et al. reported that trastuzumab upregulated PD-L1 expression through the engagement of immune effector cells and stimulation of IFN-γ secretion. However, which immune effector cells contribute to this upregulation process was unclear [89].

In gastric cancer, the preclinical study conducted by Yamashita et al. showed that trastuzumab can upregulate PD-L1 in HER2-amplified gastric cancer cells by interacting with NK cells through the secretion of IFN-γ [91].

Similar to trastuzumab, other HER-2 directed agents such as the antibody–drug conjugate Trastuzumab-deruxtecan (T-Dxd) have also shown ADCC activation effects. T-DXd has been demonstrated to increase tumor-infiltrating dendritic cells and upregulate the expression of their maturation and activation markers, increase tumor-filtered CD8+ T cells, and enhance the expression of PD-L1 and MHC class I on tumor cells [92,93].

This findings led to the hypothesis that combining trastuzumab with a second antibody that activated the host’s innate immune system (anti-PD-1/PD-L1), associated with standard cytotoxic chemotherapy, could enhance the therapeutic effects of anti-HER2 antibodies.

### 4.2. Synergistic Activity: Anti-HER2 and Anti-PD-1/PD-L1

Preclinical studies show that a monoclonal antibody against PD-1/PD-L1 can substantially boost the efficacy of anti-HER2 treatment [88,94].

The association of an anti-PD-1/PD-L1 with an anti-HER2 antibody can enhance its effect by reducing the immune escape of tumor cells, facilitating the elimination of cancer cells by trastuzumab indirectly, promoting dendritic cell trafficking and increasing the effect and the expansion of peripheral memory T cells [23,62,88,95,96,97,98].

Stagg et al. demonstrated the synergistic activity of anti-PD-1 and trastuzumab. Combining both antibodies, greater tumor regression was observed than with trastuzumab alone in a HER2-positive mouse model [88].

Junttila et al. observed that combining a trastuzumab-based bispecific HER2 antibody with anti-PD-L1 inhibited tumor growth, increasing the rates and durability of therapeutic responses [99].

Research has also been undertaken to evaluate the potential effect of combining a HER2-directed agent other than trastuzumab with immune checkpoint inhibitors.

The antibody–drug conjugate trastuzumab emtansine (T-DM1), associated with anti-CTLA4 and anti-PD-1 antibodies, improved the efficacy of the immune checkpoint blockade in immunocompetent mouse models through synergistic activation of CD8+ T cells in breast cancer [96].

Iwata et al. tested T-DXd with an anti-PD-1 antibody in mouse models, obtaining greater efficacy with the combination strategy compared to monotherapy [93].

Consistent with these findings, Zhang et al. observed a synergistic response in a preclinical study using mice that tested human PD-L1 and HER2 gene vaccinations in the treatment of HER2+ cancers [100].

Concerning the development of bispecific antibodies against PD-1/PD-L1 and HER2, diverse preclinical trials have obtained encouraging results.

A study carried out by Mittal et al. constructed a mouse bispecific antibody (BsPD-L1xrErbB2) against PD-L1 and rat HER2. This new antibody showed effective inhibition of tumor growth and increased the rate of tumor elimination in HER2-positive cell lines [101].

Gu et al. also developed a bispecific antibody against PD-1 and HER2 that showed potent blocking efficacy and strong antitumor activity both in vivo and in vitro. In addition, the antibody can cross-link HER2-positive tumor cells with T cells to form PD-1 immune synapses, directing tumor cell killing in the absence of antigen presentation [102].

Finally, Chen et al. also constructed humanized bispecific IgG1 subclass antibodies against HER2 and PD-L1, which showed stronger ADCC activity and a better anti-tumor effect than monoclonal antibodies or combination therapy in the late stage of humanized HER2-positive tumor transplantation model [103].

Based on the consistent preclinical rational, there has been growing interest in combining HER2-directed therapy with immune checkpoint inhibitors in HER2-positive gastric cancer clinical trials.

## 5. Targeting HER2 in Gastric Cancer

### 5.1. Standard Therapies

In resectable locally advanced HER2-positive tumors, the treatment of choice is perioperative chemotherapy. Since 2019, with data from the FLOT4 clinical trial (NCT01216644), chemotherapy based on the FLOT regimen (5FU, oxaliplatin, and docetaxel) constitutes the standard scheme, replacing ECF/ECX (epirubicin, cisplatin, and 5FU/capecitabine)—the previous standard perioperative treatment since the results of the MAGIC trial (NCT00002615) in 2006 [104]. In the FLOT4 trial, the median OS was 35 months (m) with ECF/ECX versus 50 months with FLOT chemotherapy (hazard ratio (HR) 0.77 (95% confidence interval (CI), 0.63–0.94); *p* = 0.012) [105].

The association of an anti-HER2 therapy with perioperative chemotherapy has not demonstrated benefit so far in this setting; therefore, it does not form part of standard treatment. However, different combinations of chemotherapy with several anti-HER2 agents are being clinically investigated.

In unresectable or metastatic disease, a platinum and fluoropyrimidine doublet chemotherapy regimen associated with trastuzumab constitutes the standard first-line treatment after the ToGa trial (NCT01041404) in 2010. Its results showed an improvement in OS in the trastuzumab-plus-chemotherapy arm (cisplatin plus capecitabine or 5-fluorouracil [5-FU]), compared with chemotherapy-alone (13.8 m vs. 11.1 m, HR 0.74 (95% CI, 0.60–0.91), *p* = 0.0046). No clinically meaningful difference in toxicity between arms was observed [50].

The choice of chemotherapy regimen is based on the patient’s general condition, comorbidities, and considering the toxicity profile of each drug. The most frequently prescribed chemotherapy combinations include cisplatin or oxaliplatin associated with 5-FU or capecitabine. There are only a few head-to-head comparisons between regimens, showing similar efficacy [106,107].

Since the addition of trastuzumab to chemotherapy, which represented a milestone in the treatment of HER2-positive disease, no further advance has been made in the first-line treatment setting.

As standard second-line treatment, paclitaxel in combination with ramucirumab demonstrated a significant improvement in OS when compared with paclitaxel alone (9.6 m vs. 7.4 m, HR 0.807 (95% CI, 0.68–0.96), *p* = 0.017), as reported in the RAINBOW trial (NCT01170663) [108].

Other second and further treatment lines include single-agent therapies such as taxanes (docetaxel in the COUGAR-02 trial (NCT00978549) [109] or paclitaxel in the WJOG4007 trial (NCT01224652) [110]), irinotecan (NCT00144378) [111], or ramucirumab in the REGARD trial (NCT00917384) [112].

Despite these treatments, the duration of clinical benefit with the current SoC is limited. The majority of patients develop treatment resistance within a year and second-line treatment options are scarce and of limited efficacy. Therefore, novel therapeutic approaches are warranted to improve survival outcomes.

### 5.2. Clinical Research in the Perioperative Setting

#### 5.2.1. Addition of Anti-HER2 Agents to Perioperative Chemotherapy

Data concerning the addition of an anti-HER2 therapy to standard perioperative chemotherapy are still scarce.

Several clinical trials are evaluating numerous combinations of chemotherapy with anti-HER2 alone or combined with immunotherapy.

One of the first studies to test this combination was the Asian phase II Trigger study (jRCTs031180006), which analyzed the combination of S1/cisplatin plus trastuzumab or placebo as preoperative treatment in 46 patients with extensive lymph node metastasis. The objective response rate (ORR) tended to be higher in the trastuzumab group than in the placebo group (66.7% vs. 36.4%, respectively; *p* = 0.08). The proportion of patients downstaging to stages 0, I or II after preoperative treatment, was also higher in the trastuzumab group (22.7% vs. 50.0%, *p* = 0.07). Survival outcomes are not available as yet [113].

Another relevant study evaluating the combination of chemotherapy with an anti-HER2 is the HER-FLOT trial (NCT01472029), a phase II study that evaluated FLOT plus trastuzumab in 56 patients. A pathological complete response (pCR) was achieved in 12 patients (21.4%) and 14 patients (25.0%) had near-complete responses. Median disease-free survival (DFS) was 42.5 months and the three-year OS rate was 82.1%. The primary endpoint (pCR > 20%) was reached. No unexpected safety issues were observed and long-term survival outcomes are promising [114].

Similarly, the phase II-III PETRARCA trial (NCT02581462) compared the standard chemotherapy regimen, FLOT, to FLOT plus trastuzumab in combination with pertuzumab. The release of negative results from the phase III JACOB trial (NCT01774786)—evaluating the addition of pertuzumab to first-line HER2-positive standard treatment—resulted in the decision to terminate enrollment [115].

Testing a different chemotherapy combination, the phase II NEOHX trial (NCT01130337), evaluated the XELOX-T regimen (capecitabine, oxaliplatin, and trastuzumab) in 36 patients. Surgery was finally performed in 31 patients, of whom 28 had R0 resection and three presented pCR. After a median follow-up of 24.1 m, the 18-month DFS was 71% (95% CI, 53–83%). An update after 102 months of follow-up showed a median OS of 79.9 months and a 60-month OS of 58% (95% CI, 40–73%) [116].

Also exploring the role of HER2-targeting in the perioperative setting, the phase II INNOVATION study (NCT02205047) randomizes patients to receive chemotherapy alone (cisplatin plus 5FU/capecitabine or FLOT), chemotherapy plus trastuzumab, or chemotherapy plus trastuzumab and pertuzumab. Results are not yet available [117].

Until new results become available, additional HER2-targeted treatment should not be recommended outside clinical trials in the perioperative setting.

#### 5.2.2. Addition of Immunotherapy Plus Anti-HER2 Agents to Perioperative Chemotherapy

Immune checkpoint inhibitors (anti-PD-1 and anti-PD-L1) are also a potential therapy under investigation in the perioperative HER2-positive setting.

In this scenario, there are several ongoing phase II studies analyzing different combinations that include immunotherapy. An Asian phase II trial (NCT04819971) is evaluating the association of tislelizumab (anti-PD-1), trastuzumab, and chemotherapy (docetaxel, S1, and oxaliplatin), while the single-arm phase II PHERFLOT (NCT02954536) will analyze pembrolizumab (anti-PD-1) in combination with trastuzumab and FLOT chemotherapy. Furthermore, an Asian phase II trial (NCT04661150) will randomize patients to receive atezolizumab (anti-PD-L1) plus trastuzumab, capecitabine, and oxaliplatin.

### 5.3. Clinical Research in the Advanced Setting

After the approval of trastuzumab in the metastatic setting, other HER2-targeted agents have been evaluated, failing to demonstrate improved efficacy compared with standard chemotherapy. Some of the main randomized clinical trials are represented in Table 1.

In spite of these unsatisfactory results, many other treatment strategies are being explored in numerous clinical trials, including chemotherapy, immunotherapy, and novel HER2-targeted therapy.

In this review, we summarize the results of the most remarkable studies and cite those that are currently ongoing.

#### 5.3.1. First Line

##### Addition of Immunotherapy to Standard First-Line HER2-Positive SoC

Exploring the role of immunotherapy in the first-line setting, an investigator-initiated single-arm phase II trial (NCT02954536) was conducted between 2016 and 2019 by Janjigian et al. [122]. Pembrolizumab associated with standard first-line therapy was evaluated in 37 patients. Chemotherapy regimens included cisplatin or oxaliplatin plus capecitabine or 5-FU. The primary endpoint—progression-free survival (PFS) after six months—was reached in 70% of patients. The ORR was 91% (32/35 patients), with six patients (17%) achieving complete response, 26 (74%) partial response, and three stable disease as best response. The median duration of response (DoR) was 10 months. After 12 months, the OS rate was 80%. The combination appeared safe, without dose-limiting toxicities.

Similarly, a phase Ib/II trial (NCT02901301) conducted by Lee et al. [123] evaluated the efficacy and safety of pembrolizumab associated with a first-line standard regimen in 43 patients. The chemotherapy regimen used was the combination of cisplatin and capecitabine. The primary endpoint was achieved, showing an ORR of 76.7% (complete responses in 14% and partial responses in 62.8% of patients). Median PFS was 8.6 m, median OS was 19.3 m, and DoR was 10.8 m. The toxicity profile was acceptable, with no patients discontinuing pembrolizumab due to severe toxicities. PD-L1 status was not related to survival.

The phase III KEYNOTE-811 study (NCT03615326) evaluated first-line SoC (trastuzumab plus chemotherapy) associated with pembrolizumab or placebo. The ORR improved by 22.7% in the pembrolizumab arm compared with the placebo group (77.4% vs. 51.9%, respectively; *p* = 0.00006). Complete responses were 11.3% in the pembrolizumab group versus 3.1% in the placebo group. The median DoR was 10.6 months for patients treated with pembrolizumab and 9.5 months for those in the placebo arm. Grade 3 or higher adverse events occurred in 57.1% of the pembrolizumab group versus 57.4% in the placebo group. These interim analyses showed that the combination of pembrolizumab with trastuzumab and chemotherapy significantly improved the objective response rate and included complete responses in some participants.

These results led to the Food and Drug Administration (FDA) accelerating the approval of pembrolizumab in combination with trastuzumab and chemotherapy in the first-line treatment for patients with HER2-positive gastric cancer. The OS and PFS results are pending [124].

In addition to pembrolizumab, other immunotherapy agents have been evaluated in this setting.

In the phase II INTEGA study (NCT03409848), the immune checkpoint inhibitors nivolumab (anti-PD-1) and ipilimumab (anti-CTLA-4) were tested in different combinations. The study assessed the combination of trastuzumab plus nivolumab plus ipilimumab in comparison with nivolumab plus trastuzumab plus FOLFOX chemotherapy. The median OS was 21.8 months in the FOLFOX arm compared with 16.4 months in the chemotherapy-free arm. The OS results obtained in the chemotherapy arm were significantly improved compared with the historic control from the ToGA trial [125].

Another novel anti-PD-1, camrelizumab, has also been tested in combination with the first-line SoC (ChiECRCT20220008). Overall, 41 patients were included and primary endpoints were ORR, disease control rate (DCR), PFS, OS, and safety. ORRs were 75% vs. 46.2% (*p* = 0.032), showing benefit for the combination arm. Survival outcomes were favorable for the camrelizumab arm for DCR (96.4 vs. 69.2%; *p* = 0.003), PFS (3.78 vs. 1.74 months; HR: 0.416, CI 0.186–0.932; *p* = 0.027), and OS (18.4 vs. 13.2 months; HR: 0.343 (CI 0.151–0.783; *p* = 0.008). The combination was well tolerated; however, higher incidences of reactive cutaneous capillary endothelial proliferation and hypothyroidism were observed in the camrelizumab arm [126].

Also evaluating camrelizumab in the first-line setting is a single-arm phase II trial (NCT05070598) analyzing its combination with pyrotinib maleate, nab-paclitaxel, and tegafur (recruiting).

In earlier-phase trials, the anti-PD-L1 HLX10 will be tested in combination with first-line SoC in a HER2-positive population (not recruiting yet) (NCT05311189).

##### Novel Anti-HER2 and Immunotherapy in First-Line HER2-Positive SoC

In the first-line scenario, novel anti-HER2 therapies are arising in combination with immunotherapy.

The phase II/III MAHOGANY trial (NCT04082364) is evaluating the efficacy of several drug combinations including margetuximab (anti-HER2 specific for the Fc domain), retifanlimab (anti-PD-1), tebotelimab (anti-PD-1/anti-LAG3 antibody), trastuzumab, and chemotherapy. The study is structured into five cohorts testing combinations. The first results of the safety analysis of 43 PD-L1-positive (CPS ≥ 1), non-MSI patients treated with margetuximab plus retifanlimab (cohort A) were presented at ESMO 2021, reporting a tumor shrinkage of 85.7% (30/35 patients). After this first safety analysis, a randomized study design will evaluate the combination of margetuximab, immunotherapy with or without chemotherapy compared to the first-line standard therapy of trastuzumab plus chemotherapy [127].

Also exploring the anti-HER2 and immunotherapy association, several ongoing studies are pending results or are in the recruitment phase.

The phase III HERIZON-GEA-01 (NCT05152147) is an ongoing trial with three treatment arms: SoC treatment with chemotherapy and trastuzumab vs. standard chemotherapy with zanidatamab (bispecific anti-HER2) vs. standard chemotherapy with zanidatamab plus tislelizumab. PD-L1 expression was not required for enrollment but will be performed retrospectively [128].

This trial was designed based on the promising results of the phase I trial in HER2-positive solid tumors where zanidatamab was evaluated. Zanidatamab was well tolerated and demonstrated an ORR > 30% in advanced gastric cancer [62,129].

A phase Ib/II Destiny-gastric03 trial (NCT04379596) investigates the efficacy of the antibody–drug conjugate against HER2, T-DXd, in several combinations including durvalumab (anti-PD-L1), pembrolizumab and chemotherapy. In the dose-escalation phase, patients with prior trastuzumab therapy received either T-DXd combined with 5-FU/capecitabine/durvalumab or capecitabine plus oxaliplatin/5-FU or capecitabine plus durvalumab. In the dose-expansion phase, therapy-naive metastatic patients are stratified by HER2 status and randomized into five study arms: T-DXd, trastuzumab plus 5-FU/capecitabine plus oxaliplatin/cisplatin, T-DXd plus 5-FU/capecitabine and oxaliplatin, T-DXd plus 5-FU/capecitabine and pembrolizumab or T-DXd plus pembrolizumab. Primary endpoints include safety, dose-finding, and ORR. Results presented at ASCO GI 2022 suggest the tolerability and feasibility of the recommended phase II doses for T-DXd plus 5-FU and T-DXd plus capecitabine. The ORR results of both arms are promising and patient recruitment is ongoing [130].

T-Dxd is an antibody–drug conjugate against HER2 with a complex mechanism of action. The antibody blocks HER2-receptor dimerization and is internalized via endocytosis. Once inside the cancer cell, the payload (cytotoxic topoisomerase I inhibitor) is released. This payload is also delivered in the extracellular matrix, acting against neighbor cancer cells (bystander effect). Moreover, it has been observed that it enhances antibody-dependent cellular cytotoxicity against the tumor. All these mechanisms are represented in Figure 3.

Finally, IBI315, the novel recombinant fully human bispecific antibody against PD-1 and HER2, is being evaluated in an exploratory single-center phase Ib/II trial (NCT05608785) as first-line treatment in a HER2-positive cohort in combination with oxaliplatin and capecitabine.

#### 5.3.2. Second and Further Lines

As for second and further treatment lines, the therapeutic options available are still limited.

Trastuzumab-deruxtecan was approved by the FDA in January 2021 as a second-line therapy based on the results of the phase II Destiny-Gastric01 study (NCT03329690). T-DXd improved responses and increased OS (median OS 12.5 versus 8.9 months) compared with physician’s choice chemotherapy (irinotecan or paclitaxel). The ORRs were 51.3% vs. 14.3%, respectively. Median DoR was also improved, with the T-DXd arm being 12.5 vs. 3.9 months. The median PFS was 5.6 vs. 3.5 months [131].

The phase III trial Destiny-Gastric04 (NCT04704934) is underway, where T-DXd will be compared with the current second-line SoC, paclitaxel plus ramucirumab.

In further treatment lines, pembrolizumab initially received accelerated approval from the FDA as a third-line-and-beyond treatment option in September 2017, based on the results of the phase II KEYNOTE-059 study (NCT02335411). Pembrolizumab was tested in monotherapy in a larger cohort of heavily pretreated patients (after two or more lines of systemic treatment). This cohort included HER2-negative and HER2-positive tumors that had previously received treatment with trastuzumab. Durable objective responses were observed in all the population, regardless of PD-L1 status, with an ORR of 12%, while a higher response rate (16%) was observed in the PD-L1-positive population [132].

Despite these data, the indication was finally withdrawn in April 2021, as the confirmatory phase III KEYNOTE-061 trial (NCT02370498) did not show any clinically meaningful improvement in OS compared to paclitaxel [133].

Several clinical trials with novel agents and combinations are ongoing. The phase II/III ASPEN-06 trial (NCT05002127) is testing evorpacept (ALX148) in combination with trastuzumab, ramucirumab, and paclitaxel; the KN026 study (NCT05427383) is evaluating the combination of KN26 (anti-HER2 agent) with chemotherapy. The phase II nextHERIZON trial (NCT05311176) is exploring the role of a vaccine against HER2 (IMU-131, named HER-Vaxx) in combination with chemotherapy or pembrolizumab. It is currently in the recruiting phase. Also recruiting is the phase II K-Umbrella trial (NCT05270889), exploring the role of tislelizumab in combination with zanidatamab as second-line therapy.

In earlier-phase trials, the single-arm phase Ib/II HER-RAM trial (NCT04888663) is evaluating ramucirumab in combination with trastuzumab and paclitaxel in patients that have progressed with a previous trastuzumab-containing chemotherapy. Furthermore, the phase Ib/II CP-MGAH22-05 trial (NCT02689284) evaluated margetuximab and demonstrated favorable results when combined with pembrolizumab, with an ORR of 24% and a disease control rate of 62% in a second-line setting.

Finally, the phase II clinical trial ILUSTRO (NCT03505320) is evaluating the novel antibody Zolbetuximab in different combinations: Zolbetuximab monotherapy or associated with immunotherapy and/or standard chemotherapy in gastric cancer patients (including HER2-positive disease) after at least two previous treatment lines. Zolbetuximab is a first-in-class monoclonal antibody that binds to the protein claudin 18.2. Targeting this protein has demonstrated promising results in HER2-negative patients, as observed in the clinical trials SPOTLIGHT (NCT03504397) [134] and GLOW (NCT03653507) [135], in which Zolbetuximab was tested in association with first-line chemotherapy. The ongoing ILUSTRO trial will provide information in pre-treated HER2-positive gastric cancer patients, with results still pending.

Although in the early stages of development, nanoparticles have become a promising strategy for diagnosis and treatment in oncology [136,137].

These molecules present higher membrane penetration and accumulate more in cancer cells. This characteristic can be used as a potential application to overcome drug resistance in pretreated patients [138].

Several preclinical studies using nanoparticles have been performed in HER2-positive gastric cancer.

HER2-directed nanoparticles associated with trastuzumab and liposomal-encapsulated doxorubicin (MM-302) showed synergistic activity in gastric and breast cancer [139].

Gold nanoparticles combined with trastuzumab demonstrated efficacy in pretreated HER2-positive gastric cancer cell lines [140].

Zhang et al. also developed a selective drug delivery system using a gold nanoshell carrier directed to HER2-positive gastric cancer cells [141].

These are only some of the studies that have been carried out with the use of nanoparticles in gastric cancer. The field of nanoparticles is an expanding area with very promising results so far in the field of oncology. However, further research and development is needed to apply these findings in clinical trials.

## 6. Conclusions

The treatment landscape of HER2-positive gastric cancer has evolved over recent years, especially with the introduction of novel anti-HER2 drugs and combination strategies. The use of immunotherapy in combination with anti-HER2 agents demonstrates a strong preclinical rationale, and clinical trials have produced encouraging results. Additionally, the diverse therapeutic drugs under development mark an exciting era in the management of HER2-positive disease.

Looking into the future, the association of immunotherapy with HER2-directed antibody–drug conjugates such as T-DXd represents a promising combination strategy, as it can further enhance the synergistic effects of both drugs. Novel antibody–drug conjugates present a particular mechanism of action, delivering a potent chemotherapy directly to target cancer cells and also the tumor cell microenvironment, which could possibly benefit the effects of immunotherapy. However, these combinational therapies need to be further explored in clinical trials.

The identification of predictive biomarkers to select accurately the patients who may benefit most from these treatments is also a challenge to overcome in future years.

A greater knowledge of the PD-1/PD-L1 pathway and the function of its different formats, as well as the optimization of its detection methods including re-assessments during the evolution of the disease, could help to optimize this selection of treatments.

Until ongoing research becomes available, it is crucial to consider treatment options within clinical trials, since these are currently patients with a poor prognosis and limited therapies available to them.

## Figures and Tables

**Figure 1 ijms-24-11403-f001:**
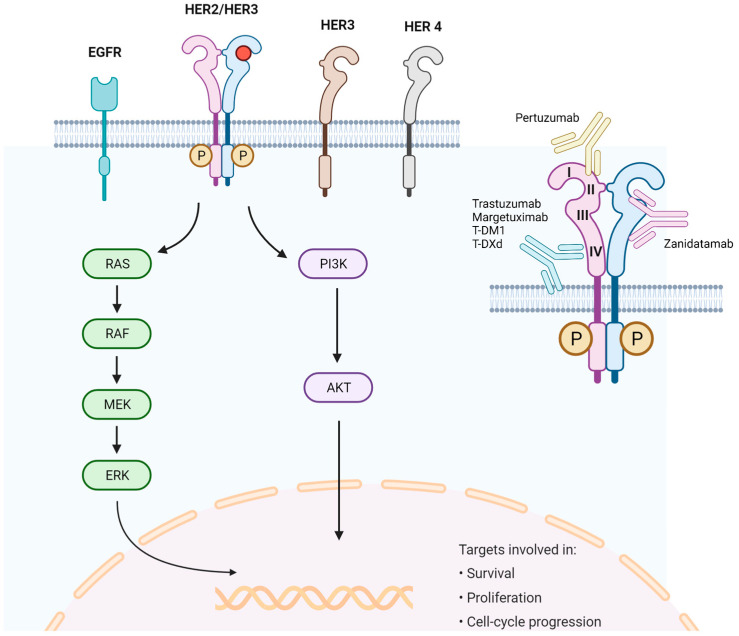
HER2 signaling pathway and binding union site of antibodies targeting HER2. HER2 receptor contains four extracellular domains: I, II, III and IV, as represented. Each HER2-directed therapy to a specific domain of the receptor.

**Figure 2 ijms-24-11403-f002:**
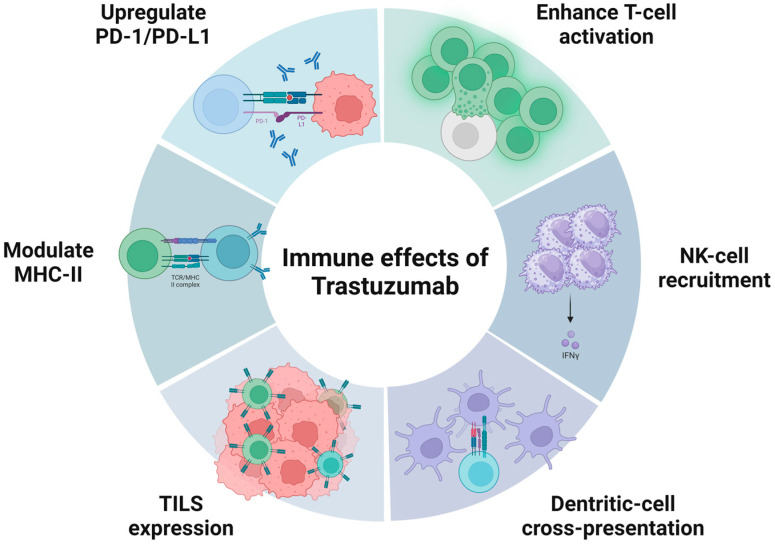
Immune effects of trastuzumab observed in preclinical studies.

**Figure 3 ijms-24-11403-f003:**
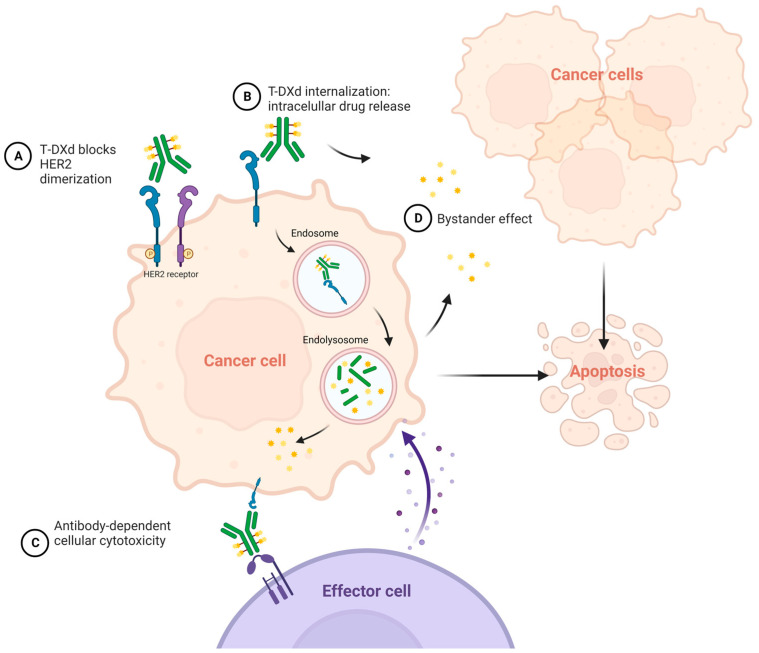
Mechanism of action of trastuzumab-deruxtecan (T-DXd).

**Table 1 ijms-24-11403-t001:** Some of the main randomized clinical trials with HER2-targeted agents in metastatic HER2-positive gastric cancer. Ref., reference; N, number of patients included; CAPOX, capecitabine plus oxaliplatin; CDDP, cisplatin; 5-FU, 5-fluorouracil; CPC, capecitabine; T-DM1, trastuzumab-emtansine, OS, overall survival; HR, hazard ratio.

Clinical Trial [Clinicaltrials.gov, accessed on 26 May 2023]	N	Line and Treatment Arms	OS (Primary Endpoint)	HR/*p*-Value
TRIO013/LOGIC (NCT00680901) Ref. [118]	487	First line CAPOX + placebo vs. CAPOX + lapatinib	10.5 vs. 12.2 m	0.91/0.34
JACOB (NCT01774786) Ref. [119]	780	First line CDDP + 5FU/CPC + trastuzumab ± pertuzumab/placebo	14.2 vs. 17.5 m	0.84/0.056
TyTAN (NCT00486954) Ref. [120]	261	Second line Paclitaxel ± lapatinib/placebo	8.9 vs. 11 m	0.84/0.1044
GATSBY (NCT01641939) Ref. [121]	415	Second line Taxane vs. T-DM1	8.6 vs. 7.9 m	1.15/0.86

## Data Availability

Not applicable since the information is gathered from published articles.

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
