# Peer review of "HER2-Positive Gastric Cancer: The Role of Immunotherapy and Novel Therapeutic Strategies"

_ijms, 2023, doi:10.3390/ijms241411403_

Round 1

Reviewer 1 Report

Title: HER2-positive gastric cancer: the role of immunotherapy and novel therapeutic strategies

 For IJMS

This concise review describes interesting topic of HER-2 positive gastric cancers. It is a good summary on this subject and will facilitate readers to understand the progress of this field. However, for a review paper (as the title indicated), it is also important to explore more into the detailed molecular mechanisms and the perspectives for the therapeutic strategies.

Therefore, it is suggested to authors to explore more details of the relationship between HER2- therapy and immunotherapy, especially the possible molecular link between HER-2 targeted therapy and PD-1/PD-L1 therapy and how combined treatment showed synergistic effect. Furthermore, the future directions of developing such therapies should be discussed.

Some other minor points:

1). PD-L1 expression and detection is often limited by just measuring the surface proteins, it also locates in cytoplasm and nuclear as well as free format in body fluids (please refer Y Wu et al Frontières in Immunology, 2019). In addition, PD-L1 can express constitutionally or can be induced by cytokines like IFN-gamma. It would be good to discuss these on page 5 third paragraph (line 206-210).

2). Some English terms or expressions can be more precise like: Page 4 line 158 was repeating line 153; page 4 line 172 needs a reference; page 4 line 177 “HER-2 positive disease” meaning gastric cancer or other cancer? Etc.

Very good, easy follow!

Author Response

Here we attach a document with the response to the review. Thank you very much. 

Reviewer 2 Report

The current manuscript is a quite interesting study on the role of immunotherapy and novel therapeutic strategies in HER2-positive gastric cancer. It is a reasonably complete review article, and addresses many crucial aspects of this disease’s pathophysiology and treatment. Nevertheless, some alterations are advised before acceptance for publication:

- In the introduction section, more should be said about gastric cancer pathophysiology, namely the different types of gastric cancer (and their characteristics and differences), symptoms, diagnosis, and societal/financial/personal impact on quality of life;

- An original figure should be produced regarding the molecular mechanisms involving HER2 in the pathogenesis and treatment of gastric cancer, for better reader understanding and visualization;

- Figure quality (image resolution) should be improved; nevertheless, they are quite illustrative, very well done;

- In what concerns novel therapies, and although clinical or even marketed preparations should be scarce, I would like the authors to mention nanosystems as an option, including some examples of successful studies; here are some references that could be included (as an example, could be others):

https://www.ncbi.nlm.nih.gov/pmc/articles/PMC9043273/

https://www.ncbi.nlm.nih.gov/pmc/articles/PMC7728832/

https://www.frontiersin.org/articles/10.3389/fonc.2022.834934/full

Author Response

We attach a document with the response to the review. Thank you very much. 
